# Denoising for 3D Point Cloud Based on Regularization of a Statistical Low-Dimensional Manifold

**DOI:** 10.3390/s22072666

**Published:** 2022-03-30

**Authors:** Youyu Liu, Baozhu Zou, Jiao Xu, Siyang Yang, Yi Li

**Affiliations:** 1Key Laboratory of Advanced Perception and Intelligent Control of High-End Equipment, Ministry of Education, Wuhu 241000, China; liuyyu@ahpu.edu.cn (Y.L.); 2012124@ahpu.edu.cn (S.Y.); 2210110112@stu.ahpu.edu.cn (Y.L.); 2School of Mechanical Engineering, Anhui Polytechnic University, Wuhu 241000, China; 3System Office, Wuhu Changxin Technology Co., Ltd., Wuhu 241009, China; jxu@tokengroup.com

**Keywords:** point cloud denoising, statistical low-dimensional manifold, regularization, Gaussian noise, Laplace noise

## Abstract

A point cloud obtained by stereo matching algorithm or three-dimensional (3D) scanner generally contains much complex noise, which will affect the accuracy of subsequent surface reconstruction or visualization processing. To eliminate the complex noise, a new regularization algorithm for denoising was proposed. In view of the fact that 3D point clouds have low-dimensional structures, a statistical low-dimensional manifold (SLDM) model was established. By regularizing its dimensions, the denoising problem of the point cloud was expressed as an optimization problem based on the geometric constraints of the regularization term of the manifold. A low-dimensional smooth manifold model was constructed by discrete sampling, and solved by means of a statistical method and an alternating iterative method. The performance of the denoising algorithm was quantitatively evaluated from three aspects, i.e., the signal-to-noise ratio (SNR), mean square error (MSE) and structural similarity (SSIM). Analysis and comparison of performance showed that compared with the algebraic point-set surface (APSS), non-local denoising (NLD) and feature graph learning (FGL) algorithms, the mean SNR of the point cloud denoised using the proposed method increased by 1.22 DB, 1.81 DB and 1.20 DB, respectively, its mean MSE decreased by 0.096, 0.086 and 0.076, respectively, and its mean SSIM decreased by 0.023, 0.022 and 0.020, respectively, which shows that the proposed method is more effective in eliminating Gaussian noise and Laplace noise in common point clouds. The application cases showed that the proposed algorithm can retain the geometric feature information of point clouds while eliminating complex noise.

## 1. Introduction

Due to their simplicity, flexibility and strong representative ability, three-dimensional (3D) point clouds are more and more widely used in many fields [1,2,3], such as object recognition and surface reconstruction. Point cloud data are mainly obtained in two ways [4,5], namely using a stereo matching algorithm or a 3D scanner. The former obtains the point cloud by projecting all pixels into the 3D space through matching [6]. Due to the fuzzy matching, a lot of noise will be present in the obtained point cloud. The latter, including laser scanners [7], structured light scanners [8] and lidar [9], can quickly obtain complete point clouds. However, due to the uncertainties of measurement errors, reflectance from objects, occlusion, illumination and the environment, the obtained point cloud data of objects often contain a large number of complex noise points [10]. Noise not only deforms the bottom manifold structure of point clouds [11], which is not conducive to their surface reconstruction and visualization, but also adds useless information [12], and then reduces the accuracy of the extraction of their features.

Point cloud denoising has become one of the hot fields of 3D geometric data processing. Zeng et al. [13] extended a previously proposed graph Laplace regularization (GLR), which uses the patch manifold prior to seeking self-similar patches in order to fulfill the purpose of point cloud denoising. Dinesh et al. [14] proposed a fast graph-based local algorithm, but it was found that the graph signal denoising algorithm would lead to the problems of clustering and deformation. Duan et al. [15] proposed a weighted multi-projection (WMP) algorithm to estimate a local tangent plane at each 3D point, and then reconstructed each 3D point through the weighted averaging of its projections on multiple tangent planes, so as to alleviate the problems of clustering and deformation. Wei et al. [16] applied the feature graph learning (FGL) algorithm to point cloud denoising. Different from setting the edge weight parameters manually, when the available signal is smooth relative to the graph, the edge weight parameters are strictly optimized through feature measurement learning, so as to realize the accurate and fast denoising of the point cloud. The above algorithms are derived from formulae that explicitly assume Gaussian noise, and have achieved good results regarding the additional noise generated on the object surface, but theoretically, they are not suitable for the actual complex noise far away from the object noise points. The removal of actual complex noise points is still challenging.

To solve the issue of the actual complex noise of point clouds in objects, a denoising approach for 3D point clouds, based on the regularization of the statistical low-dimensional manifold (R-SLDM), is proposed in this article. Firstly, the slight and non-sparse noise (such as Gaussian noise) and large and sparse noise (such as Laplace noise) (see Figure 1) were combined into actual complex noise. Secondly, in view of the fact that 3D point clouds have low-dimensional structures, the point cloud model with noise was distributed on the low-dimensional manifold in a high-dimensional space. Moreover, the regularization term of the manifold was used for the denoising of prior information to maintain the basic structure embedded in the low-dimensional manifold. Finally, the denoising problem of point clouds was expressed as an optimization problem based on the geometric constraints of the regularization term of the manifold, which was approximately solved by means of a statistical method and an alternating iterative method.

The main contributions of our work are as follows:(1)A denoising algorithm was designed for Gaussian noise and Laplace noise to denoise the two kinds of noise together;(2)Using the regularization term of the manifold and the fidelity term of the noise, the basic structure of a denoised point cloud was maintained;(3)Discrete sampling was used to construct low-dimensional manifolds to avoid a large number of calculations.

## 2. Related Works

At present, there are two kinds of denoising methods for point cloud data: local methods [14,17,18,19,20,21,22,23,24,25] and non-local methods [13,15,16,26,27,28]. The former involves the denoising of the point cloud based on the neighborhood of points, while the latter involves identifying similar point patches from the local neighborhood and combining this group of patches for denoising.

**(1) Local Method**. Alexa et al. [17] defined a point cloud surface by means of moving least-squares projection, projected the noise points onto the least-squares surface, and then smoothed the data. Guennebaud et al. [18] and Öztireli et al. [19] proposed the algebraic point set surface (APSS) and robust implicit moving least-squares (RIMLS), respectively, for denoising based on moving least-squares (MLS), but it was found that over-smoothing could easily be produced [20,21]. Lipman et al. [22] introduced local optimal projection (LOP) in point cloud denoising, of which the basic principle is to project the point set to the natural surface of the point cloud to reduce noise. However, the algorithm has a limited effect in terms of preserving the features of the point cloud. Huang et al. [23] proposed a weighted local optimal projection (WLOP) algorithm based on the local optimal projection algorithm, which can retain the features of a point cloud, but it is time-consuming. Moreover, they also proposed an anisotropic WLOP (AWLOP) algorithm by modifying WLOP, and used anisotropic weighting functions to retain the sharp features of the point cloud model [24]; however, it can easily produce additional features [20,21]. Dinesh et al. [25] solved different types of additive noise by using two fidelity terms through feature graph Laplace regularization (FGLR).

**(2) Non-Local Method**. Dabov et al. [26] and Rosman et al. [27] spread image denoising algorithms such as non-local mean (NLM) and block-matching 3D Denoising (BM3D) to point cloud denoising. However, this was found to depend on the self-similarity among the surface blocks in the point cloud, and required a large amount of calculation. Deschaud et al. [28] used the polynomial coefficients of local MLS surfaces as neighborhood descriptors to calculate the similarity of points, and then proposed a non-local denoising (NLD) algorithm. Zeng et al. [13] proposed the GLR denoising algorithm to seek self-similar patches to denoise the point cloud. Different from the direct smoothing of 3D coordinate points, Duan et al. [15] estimated the local tangent plane of each point based on the graphical method, and realized point cloud denoising through weighted projection. Wei et al. [16] used the FGL algorithm for point cloud denoising.

The method proposed in this article is categorized as a non-local method. Similar to [13,16,25], we also used regularization for denoising. However, the methods given in references [13,16] involve the establishing of prior information based on Gaussian noise, which is not suitable for complex information including both Laplace noise and Gaussian noise, so it is difficult to apply these methods in practical denoising. On the other hand, although two kinds of additive noise are studied in reference [25], two different algorithms are designed for them; thus, the two kinds of noise are not solved uniformly. Our proposed method is intended to solve the above two problems.

## 3. Methods

### 3.1. Low-Dimensional Manifold Model

Image u∈ℝm×n (indicating that u is in the number field with the size of m×n); 𝒫(u)(x) is defined as a two-dimensional (2D) rectangular pixel block of the u, with a size of s1×s2. Pixel x is in the upper left corner of the rectangle, and ∀ x∈Ω¯={1,2,…,m}×{1,2,…,n}. The set of all pixel blocks is called the pixel block set of the **u**, which is represented by P(u) [29]:(1)P(u)={𝒫(u)(x):x∈Ω¯}⊂ℝd, d=s1×s2

For the image u, a point cloud in ℝd is given from the P(u), which is close to a smooth manifold embedded in ℝd. This potential smooth manifold is called a block manifold; it is associated with u, and is denoted as ℳ(u). Based on the fact that there are often low-dimensional structures in many natural image block manifolds [29,30], Osher et al. designed a low-dimensional manifold model [31], as shown in Equation (2):(2)argminu∈ℝm×nℳ⊂ℝddim(ℳ), subject to: b=Φu+ε, P(u)⊂ℳ

Moreover, dim(ℳ) in Equation (2) is expressed as follows: (3)dim(ℳ)=∑i=1d||∇ℳαi(x)||L2(ℳ)2
where αi(x)=xi; ∀ x=(x1,⋯,xd)∈ℳ⊂ℝd

An image restoration model for low-dimensional manifolds is obtained from Equations (2) and (3), as shown in Equation (4):
(4)argminu∈ℝm×nℳ⊂ℝd∑i=1d||∇ℳαi(x)||L2(ℳ)2+μ∥b−Φu||22, subject to: P(u)⊂ℳ

### 3.2. Point Cloud Denoising Model

#### 3.2.1. 3D Point Cloud and Noise Model

Point cloud N={ni}i=1N, and ni∈ℝ3. Let N=[n1,…,nN]⊤∈ℝN×3, and then its observation model can be expressed as Equation (5):(5)N=U+G+L
where U, G, L∈ℝN×3.

Our goal in developing this model was to restore the observation model N to the ideal model U under the interference of Gaussian noise G and Laplace noise L.

#### 3.2.2. Statistical Low-Dimensional Manifold Model

A surface block {nm}m=1M⊂N is defined in the point cloud, from which M subsets are selected as the block centers. The block set pm centered on nm is defined as the set of k points closest to nm, and U𝑀m=1pm=𝒩. Let pm∈ℝ3k be the block coordinates, composed of k points in pm.

A low-dimensional smooth manifold, in which pm samples are embedded in ℝ3k, is represented by ℳ(U). The designed statistical low-dimensional manifold (SLDM) model is shown in Equation (6):
(6)arg minU dim(ℳ(U)), subject to:N=U+G+L, pm⊂ℳ(U)

The maximum a posteriori (MAP) problem can be described by a regularization term and a fidelity term, as follows:
(7)$$arg\;\underset{U}{min}\;\dim\left(ℳ(U)\right)+𝜆∥N-U-L{∥}_{F}^{2},\;\mathrm{subject}\;\mathrm{to}:\;p_{m}\subset ℳ\left(U\right)$$
where $∥N-U-L{∥}_{F}^{2}$ is the fidelity item, which is used to ensure the similarity before and after point cloud processing; $\underset{U}{min}\dim\left(ℳ(U)\right)$ is the regularization term to enhance the output;
λ is a parameter to weigh the relationship between the fidelity term and the regularity term.

Blocks are usually not a single smooth manifold, which may have different dimensions and correspond to different patterns of images [29]; thus, dim(ℳ(U)) of the block manifold is a function of ℳ(U). The integral of dim(ℳ(U)) on ℳ is used for regularization, as shown in Equation (8):(8)arg minU∫ℳdim(ℳ(U))(p)dp+𝜆∥N−U−L∥F2, subject to: pm⊂ℳ(U)
where p∈ℝ3k is a point on ℳ; dim(ℳ(U))(p) is the manifold size of ℳ(U) at p.

According to Equation (3),(9)$$\dim\left(ℳ\right)\left(p\right)=\sum_{i=1}^{3k}{||\nabla_{ℳ}f_{i}\left(p\right)||}_{F}^{2}$$

Substituting Equation (9) into Equation (8),
(10)arg minU∑i=13k∫ℳ||∇ℳfi(p)||2dp+λ∥N−U−L∥F2, subject to:N=U+G+L
where ∀p=[p1,…,p3k]⊤∈ℳ⊂ℝ3k, fi(p)=pi; ∇ℳfi(p) is the gradient of function fi at p on ℳ.

### 3.3. Solution of Point Cloud Denoising Model

#### 3.3.1. Solution Principle

Manifold ℳ is discretely sampled to construct a discrete graph G, and to describe the low-dimensional smooth manifold model. Its vertex set is a visible surface block 𝒫={pm}m=1M, and pm∈ℳ(U)⊂ℝ3k, in which the edge weight of the m and n blocks are defined as follows:(11)wm,n={(ρmρn)−1/γexp(−∥pm−pn∥222ϵ2)    ∥pm−pn∥2<r0                            otherwise
where (ρmρn)−1/γ is a normalized term; ρn=Σm=1Mψ(||pm−pn||2) is the expression before normalization; ψ(⋅) is the weight kernel function. Under these conditions, an r-neighborhood graph is constructed with no edge being greater than r; r=ϵCr.

A symmetric adjacency matrix W∈ℝM×M is defined by using the edge weight in Equation (11) to represent graph G, in which W(m,n)**=** W(n,m)**=** wm,n. Moreover, D(m,m)=∑nwm,n. According to the literature [32], the Laplacian matrix of the combined graph is defined as M=D−W. For the regularization term in Equation (10), fi is sampled at the position of 𝒫 to obtain the discrete form fi=[fi(p1)…fi(pM)]⊤. The regularizer fi⊤Mfi [33] is derived from M, as shown in Equation (12).
(12)fi⊤Mfi=∑(m,n)∈𝜏wm,n(fi(pm)−fi(pn))2

According to studies [34,35],
(13)$$\underset{\begin{array}{c} M\to \infty \\ \epsilon \to 0,\delta \to 0 \end{array}}{lim}f_{i}^{⊤}Mf_{i}∼\frac{1}{\left|ℳ\right|}\int_{ℳ}∥\nabla_{ℳ}f_{i}{\left(p\right)∥}_{2}^{2}dp$$

With the increasing of sample number M and the decreasing of neighborhood radius r, $f_{i}^{⊤}Mf_{i}$ is close to its limit of smoothness. If the manifold dimension δ is low, even if the block manifold is embedded in a high-dimensional space, it can reasonably approximate the continuous regularization function [36].

According to Equations (9) and (13),(14)$$\underset{\begin{array}{c} M\to \infty \\ \epsilon \to 0,\delta \to 0 \end{array}}{lim}\left|ℳ\right|\sum_{i=1}^{3k}f_{i}^{⊤}Mf_{i}∼\sum_{i=1}^{3k}\int_{ℳ}∥\nabla_{ℳ}f_{i}{\left(p\right)∥}_{2}^{2}dp$$

In order to define $f_{i}$, the ${\left\{p_{m}\right\}}_{m=1}^{M}$ must be sorted. The following Equation (15) can be inferred from Equation (12):
(15)$$\sum_{i=1}^{3k}f_{i}^{⊤}Mf_{i}=\sum_{(m,n)\in 𝜏}w_{m,n}∥p_{m}-f_{i}{p_{n}∥}_{2}^{2}$$
where $∥p_{m}-{p_{n}∥}_{2}$ can determine the block similarity. It can be seen from Equation (15) that the regularization does not need the coordinate function $f_{i}$; thus, the surface blocks do not need to be sorted.

According to the constructed discrete graph G, the global graph Laplace matrix Mp=∑i=13kSTMS∈ℝkM×N can be obtained, in which kM is the number of points of all surface blocks, and S∈{0,1}kM×N. Therefore, Equation (10) can be transformed into Equation (16):
(16)arg minU(PxTMpPx+PyTMpPy+PzTMpPz)+λ∥N−U−L∥F2, subject to:N=U+G+L where Px, Py, Pz∈ℝkM.

Let P=[Px,Py,Pz]∈ℝkM×3, then,
(17)tr(P⊤MpP)=PxTMpPx+PyTMpPy+PzTMpPz
where P is related to the denoised point cloud sample, and P=SU−C, C∈ℝkM×3.

The objective function (16) can be rewritten as shown in Equation (18):(18)$$arg\;\underset{U}{min}\mathrm{tr}\;\left({\left(SU-C\right)}^{⊤}M_{p}\left(SU-C\right)\right)+\lambda ∥N-U-L∥{}_{F}^{2},\;\mathrm{subject}\;\mathrm{to}:N=U+G+L$$

For any point $n_{i}$ in the point cloud N, let $S_{i}$ represent the average distance from this point to k points in the neighborhood, and then the distance threshold is obtained, as shown in Equation (19):
(19)$$d_{\mathrm{threshold}}=\frac{\sum_{i=1}^{n}S_{i}}{N}\pm std\cdot \frac{\sum_{i=1}^{n}{\left(S_{i}-\frac{\sum_{i=1}^{n}S_{i}}{N}\right)}^{2}}{N}$$

The point with a distance between $n_{i}$ and a neighborhood outside $\left(\frac{\sum_{i=1}^{n}S_{i}}{N}-std\cdot \frac{\sum_{i=1}^{n}{\left(S_{i}-\frac{\sum_{i=1}^{n}S_{i}}{N}\right)}^{2}}{N},\frac{\sum_{i=1}^{n}S_{i}}{N}+std\cdot \frac{\sum_{i=1}^{n}{\left(S_{i}-\frac{\sum_{i=1}^{n}S_{i}}{N}\right)}^{2}}{N}\right)$
is Laplace noise. Therefore, the position matrix L of Laplace noise can be obtained using a statistical method.

To solve Equation (18) approximately, an alternating optimization method is adopted: in the loop, fix $M_{p}$ and update U, then update $M_{p}$ given U, and repeat until convergence. In each iteration, $M_{p}$ is updated according to the above method. In order to fix $M_{p}$ and optimize U, the coordinates (x,y,z) of each point are given by Equation (20):(20)$$\left(S^{T}M_{p}S+\lambda E_{q}\right)U_{q}=\lambda \left(N_{q}+L_{q}\right)+S^{T}M_{p}C_{q}$$
where $E_{q}$ is an identity matrix with the same size as $M_{p}$; U is solved by means of an iterative method until the result converges.

#### 3.3.2. Algorithm Design

Based on the solution principle, the denoising process for the 3D point cloud, based on the regularization of SLDM (R-SLDM), is shown in Figure 2, and its algorithm (Algorithm 1) is shown as follows:
**Algorithm 1**: Denoising for 3D point cloud based on R-**Input**: N, k, s, λ, ξ.**Output**: Denoised cloud U.1: Initializing U with N;2: **for** iter = 1, 2, … r **do**;3: Sampling s points from U as a block center;4: Find the k nearest neighbors of the center of each block to form a surface block;5: Optimizing objective function: arg minUtr ((SU−C)⊤Mp(SU−C))+λ∥N−U−L∥F2, subject to: N=U+G+L;6: L←dthreshold=∑i=1nSiN±std⋅∑i=1n(Si−∑i=1nSiN)2N;
7:MP←∑i=13kSTMS;
8: Nq←(STMpS+λEq)Uq=λ(Nq+Lq)+STMpCq; q∈{x,y,z};9: U converges;10: **end for**

#### 3.3.3. Performance Evaluation

The performance of the denoising algorithm can be quantitatively evaluated based on the signal-to-noise ratio (SNR), mean square error (MSE) and structural similarity (SSIM).

The SNR [13] is shown in Equation (21):
(21)$$\mathrm{SNR}\left(U_{1},U_{2}\right)10\mathrm{lg}\frac{\frac{2}{N_{2}}\sum_{u_{i}\in U_{2}}{∥u_{j}∥}_{2}^{2}}{\frac{1}{N_{1}}\sum_{u_{i}\in U_{1}}\underset{u_{j}\in U_{2}}{min}{∥u_{i}-u_{j}∥}_{2}^{2}+\frac{1}{N_{2}}\sum_{u_{i}\in U_{2}}\underset{u_{j}\in U_{1}}{min}{∥u_{i}-u_{j}∥}_{2}^{2}}$$


The MSE [15] is shown in Equation (22):
(22)MSE(U1,U2)12(1N1∑ui∈U1minuj∈U2∥ui−uj∥22+1N2∑ui∈U2minuj∈U1∥ui−uj∥22)
where U1={ui}i=1N1, ui∈ℝ3; U2={uj}j=1N2, uj∈ℝ3.

The SSIM [37] is shown in Equation (23):(23)$$\mathrm{SSIM}\left(U_{1},U_{2}\right)=\frac{\left(2\mu_{1}\mu_{2}+C_{1}\right)\left(2\sigma_{12}+C_{2}\right)}{\left(\mu_{1}^{2}+\mu_{2}^{2}+C_{1}\right)\left(\sigma_{1}^{2}+\sigma_{2}^{2}+C_{2}\right)}$$
where $\mu_{1}=\sum_{u_{i}\in U_{1}}w_{i}|u_{i}|$, $\mu_{2}=\sum_{u_{j}\in U_{2}}w_{j}|u_{j}|$, $\sigma_{1}={\left[\sum_{u_{i}\in U_{1}}w_{i}{(|u_{i}-\mu_{1}|)}^{2}\right]}^{\frac{1}{2}}$, $\sigma_{2}={\left[\sum_{u_{j}\in U_{1}}w_{j}{(|u_{j}-\mu_{1}|)}^{2}\right]}^{\frac{1}{2}}$, $\sigma_{12}=\frac{1}{2}[\sum_{u_{i}\in U_{1}}w_{i}(|u_{i}-\mu_{1}|)(|u_{j}|-\mu_{2})+\sum_{u_{j}\in U_{2}}w_{j}(|u_{i}|-\mu_{1})(|u_{j}|-\mu_{2})]$,
$\overset{N}{\underset{i=1}{\sum }}w_{i}=1$.

With the SNR, MSE and SSIM, the difference between the ground-truth point cloud and the denoised one can be reasonably compared. The larger the SNR and SSIM, the lower the distortion of point cloud denoising, and the better the denoising effect. The smaller the MSE value, the lower the deviation between the ground-truth value and the denoised cloud, and the better the denoising effect.

## 4. Results and Discussion

### 4.1. Comparative Analysis of Denoising Performance

To verify the effectiveness of the proposed algorithm, the public point cloud model known as Daratech was used to carry out the denoising research, and the performance was compared with the existing algorithms under different noise intensity conditions.

As shown in Figure 3a, Daratech’s ground-truth point cloud model contains 32,003 data points. The Gaussian noise and Laplacian noise, with a mean value of 0 and standard deviations (σ) of 0.1, 0.2, 0.3, 0.4, 0.5, were added to the model of the ground-truth point cloud, for which the noise models with σ = 0.2 and σ = 0.4 are shown in Figure 4. To conveniently display the effect and denoising results after adding noise, color information was added to the resulting model, and the color bar represents depth information. The R-SLDM proposed in this article was used to denoise the 3D point cloud model under different noise intensities, as shown in Figure 4, in which k is 16, s is 50%, λ is 21 and ξ is 15. From the denoising results in Figure 3b,c, it can be seen that the denoising algorithm proposed in this article can effectively eliminate Gaussian noise and Laplace noise in the common point cloud, and can properly retain their original geometric features. Thus, it is an effective point cloud denoising algorithm.

The point cloud models under different noise intensities, as shown in Figure 4, were denoised by APSS, NLD and FLD, respectively. It can be seen from Figure 5 that the color distribution on the surface of the point cloud model was uneven, and thus, did not show the plane characteristics well. Moreover, there was noise far away from the main body. Compared with Figure 5, the color distribution on the surfaces shown in Figure 6 and Figure 7 was relatively uniform, but there was also noise far away from the main body. It can be seen from the results of Figure 5, Figure 6 and Figure 7 that the Laplace noise could not be well eliminated. As can be seen from Table 1, for the five kinds of noise levels with different σ values (0.01 to 0.05), the mean SNRs of APSS and NLD after denoising were 46.79 DB and 46.42 DB, respectively, with little difference. The mean SNR of the latest FGL method after denoising was 47.03 DB, while that of the proposed algorithm in this article was 48.23 DB, which was 1.22 DB more than that of APSS, 1.81 DB more than that of NLD and 1.20 DB more than that of FGL. It was thus better than the latter three. The distortion of the denoising process of the proposed algorithm was the lowest. It can be seen from Table 2 that the mean MSE obtained by the R-SLDM in this article was as low as 0.159, 0.096 lower than that of APSS, 0.086 lower than that of NLD, and 0.076 lower than that of FLD. The deviation between the ground-truth value and the denoised cloud was the lowest. It can be seen from Table 3 that the mean SSIM with noise was 0.906, and those of APSS, WLOP and FGL were 0.944, 0.945 and 0.947, respectively, with all of these playing a certain role in the composite noise. However, the mean SSIM of the algorithm in this article was 0.967, which was 0.023, 0.022 and 0.020 more than that of the other three algorithms. Comparative experiments showed that the proposed algorithm is more suitable for denoising noise in practice.

### 4.2. Denoising Application for 3D Point Cloud

To further verify the proposed R-SLDM model, a self-built laser-scanning platform, as shown in Figure 8, was used to collect the point cloud of the four objects with noise, and the algorithm in this article was used for denoising on a workstation. The workstation parameters were as follows: the basic frequency was 3.90 GHz; the CPU model was AMD Ryzen73800X; and the graphics card model was 3060.

The objects shown in Figure 9 were scanned by laser, and the obtained 3D point cloud is shown in Figure 10. To clearly illustrate the point cloud noise of objects and the denoising performance of various algorithms, we locally amplified the scanned point cloud and denoising results. It can be seen from the local enlarged view of the point clouds of the four objects in Figure 10 that there were regular noises close to the main bodies and irregular noises far away from them in the point clouds obtained by the laser-scanning platform. The results obtained for the denoising of the point cloud with the APSS algorithm are shown in Figure 11. Taking object 3 as an example, it is obvious from Figure 10c and Figure 11c that there was little difference before and after point cloud denoising. Consequently, the APSS algorithm could not eliminate the noise far away from the main body, and the removal effect of noise on the object surface was also not effective. The results obtained for the denoising of the point cloud shown in Figure 10 using the NLD algorithm are shown in Figure 12. As can be seen from Figure 12a,b, the noise on the object surface was reduced considerably, but the noise far away from the main body still existed. As can be seen from Figure 12c,d, noise still existed on the surface of the workpiece and away from the main body. By analyzing the structural information of the four objects, it can be seen that the structures of objects 1 and 2 were simple with regular planes or surfaces, and those of objects 3 and 4 were complex, containing a large number of irregular holes and surfaces. It was found that the NLD algorithm was not suitable for the point clouds with complex structures and that were far away from the main body. The denoising results obtained with the FGL algorithm are shown in Figure 13. The noise on the surface of the four objects was well eliminated, and the noise far away from the main body was not removed. The results obtained for the denoising of the point cloud shown in Figure 10 using the proposed R-SLDM algorithm are shown in Figure 14, in which k is 16, s is 50%, λ is 21 and ξ is 15. The results show that the surfaces of the objects were smooth and the noise away from the main body became less. Compared with APSS, NLD and FGL, the R-SLDM algorithm proposed in this article was able to effectively remove large-scale irregular noise far from the main body and small-scale regular noise close to the main body. Taking object 1 in Table 4 as an example, the SNR value of the R-SLDM algorithm was 47.83 db, which was 2.62 DB higher than that of APSS, 1.08 DB higher than that of NLD and 0.91 DB higher than that of FLD. The SNR value of the proposed R-SLDM algorithm was higher than that of APSS, NLD and FGL. For different objects, the mean SNR of R-SLDM was 48.49 DB, which was 1.02 DB higher than that of APSS, 1.14 DB higher than that of NLD and 0.94 DB higher than that of FGL. From Table 5 it can be inferred that the mean MSE of the proposed R-SLDM algorithm was reduced to 3.165 × 10^−3^, a lower value than that of APSS (1.213 × 10^−3^), NLD (1.107 × 10^−3^) and FGL (0.451 × 10^−3^). According to Table 6, the mean SSIM values of APSS, WLOP and FGL were 0.935, 0.937 and 0.950, respectively. All of the three algorithms maintained the structure of the point cloud. Nevertheless, the mean SSIM of the proposed algorithm was 0.973, which was 0.038, 0.036 and 0.023 more than that of the other three algorithms, and it retained the real structure of the point cloud more completely. Comparative experiments showed that the R-SLDM algorithm not only eliminated complex noise, but also effectively retained the geometric information of the point cloud, and the distortion rate was low.

## 5. Conclusions

A SLDM model was established in this paper, which was sampled discretely. The position matrix of the ideal point cloud was approximately calculated using the statistical method and an alternating iterative method; then, the denoising for the 3D point cloud, based on R-SLDM, which can be widely applied to the denoising of point clouds with significant geometric characteristics, was determined. Different algorithms were used to denoise the Daratech model with five kinds of noise levels, and SNR, MES and SSIM were used for comparative analysis. Compared with APSS, NLD and FGL, the mean SNR of the point cloud denoised by the proposed algorithm increased by 1.22 DB, 1.81 DB and 1.20 DB, respectively; the mean MSE decreased by 0.096, 0.086 and 0.076, respectively; the mean SSIM increased by 0.023, 0.022 and 0.020, respectively. Different algorithms were used to denoise the 3D noisy point clouds of the four kinds of objects, which shows that compared with APSS, NLD and FGL, the mean SNR of the point cloud denoised by the proposed algorithm increased by 2.62 DB, 1.08 DB and 0.91 DB, respectively; the mean MSE decreased by 1.213 × 10^−3^, 1.107 × 10^−3^ and 0.451 × 10^−3^, respectively; and the mean SSIM increased by 0.038, 0.036 and 0.023, respectively. The results show that the proposed method is more suitable for denoising practical complex noise than the existing algorithms, and can clearly retain the salient features of the objects without excessive smoothing. The method proposed in this article has achieved great success in static point cloud denoising, but has not been verified in the field of dynamic point clouds. In future work, we will consider the noise caused by irregular sampling and the change of the number of points in each frame of the dynamic point cloud. The dynamic point cloud sequence can be divided into each frame independently, and then the proposed method can be extended to the field of dynamic point clouds with strengthening of the inter-frame correlation.

## Figures and Tables

**Figure 1 sensors-22-02666-f001:**
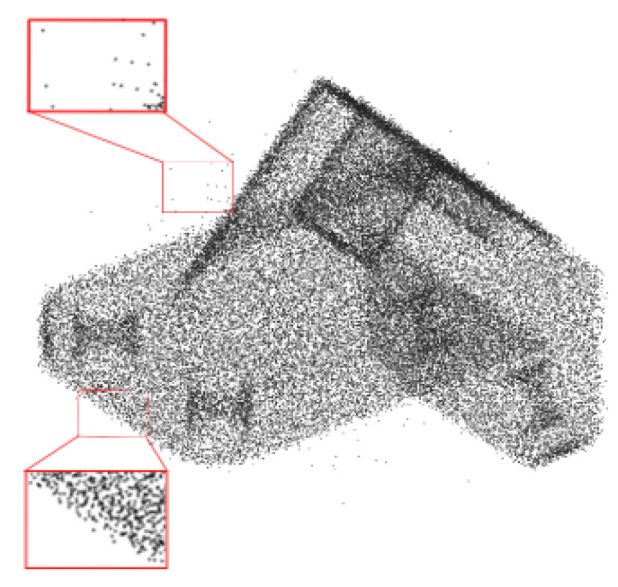
Point cloud model with Laplace noise and Gaussian noise.

**Figure 2 sensors-22-02666-f002:**
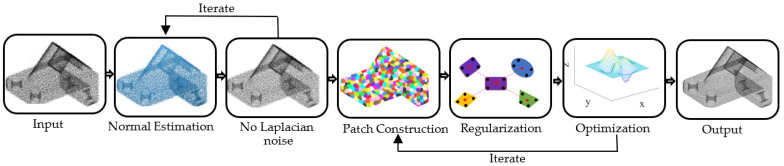
The flowchart of the proposed point cloud denoising algorithm.

**Figure 3 sensors-22-02666-f003:**
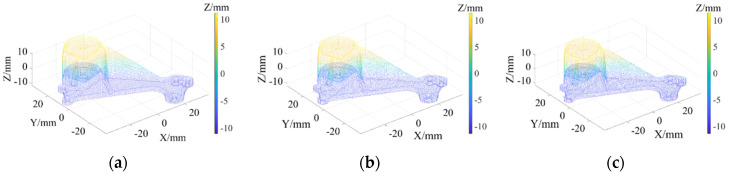
Ground-truth point cloud and the denoising results by R-SLDM under different noise intensities: (**a**) ground-truth; (**b**) R-SLDM (σ=0.2); (**c**) R-SLDM (σ=0.4 ).

**Figure 4 sensors-22-02666-f004:**
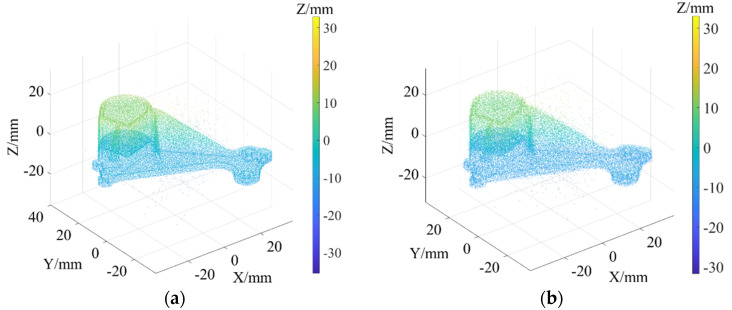
Point cloud models under different noise intensities: (**a**) noise (σ=0.2); (**b**) noise (σ=0.4).

**Figure 5 sensors-22-02666-f005:**
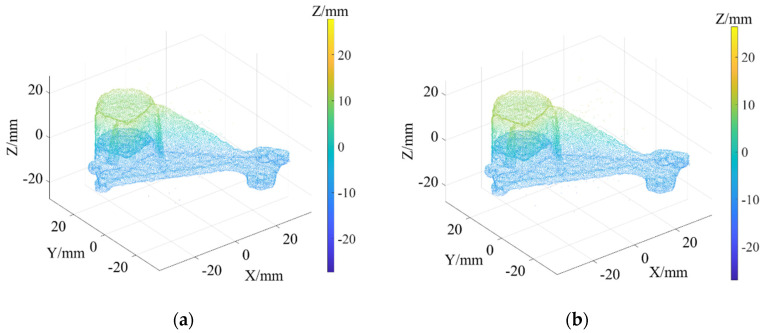
Denoising results of APSS under different noise intensities: (**a**) APSS (σ=0.2); (**b**) APSS (σ=0.4).

**Figure 6 sensors-22-02666-f006:**
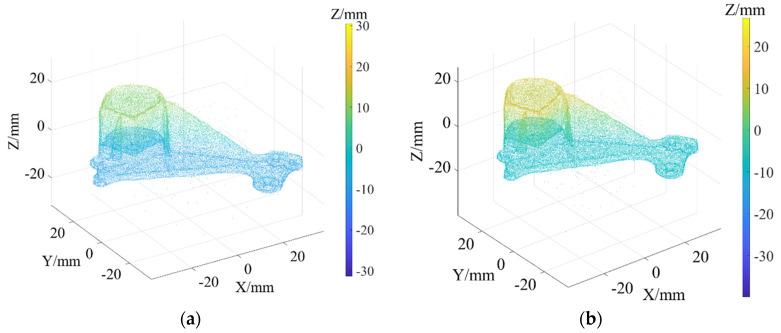
Denoising results of NLD under different noise intensities: (**a**) NLD (σ=0.2); (**b**) NLD (σ=0.4).

**Figure 7 sensors-22-02666-f007:**
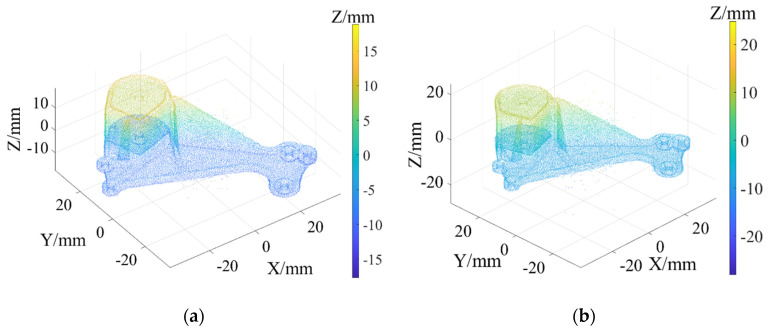
Denoising results of FGL under different noise intensities: (**a**) FGL (σ=0.2); (**b**) FGL (σ=0.4).

**Figure 8 sensors-22-02666-f008:**
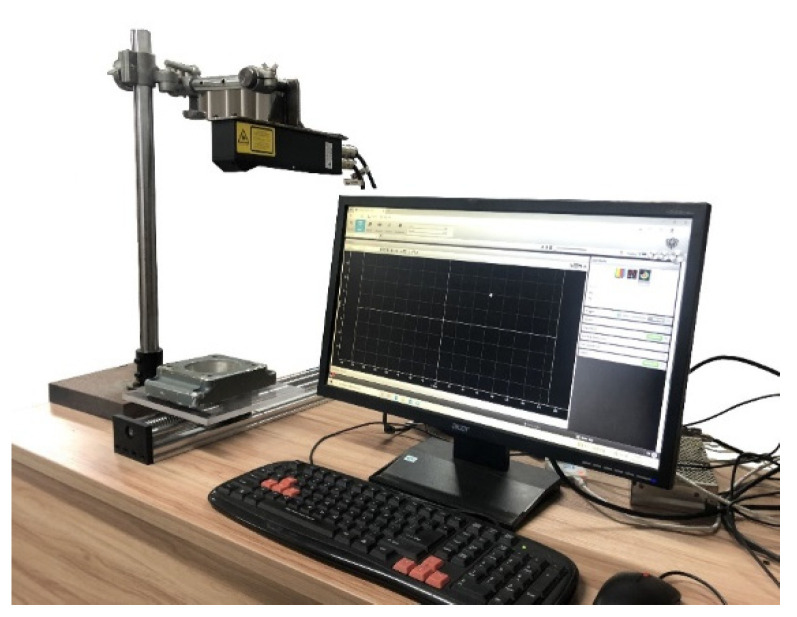
Laser-scanning platform.

**Figure 9 sensors-22-02666-f009:**
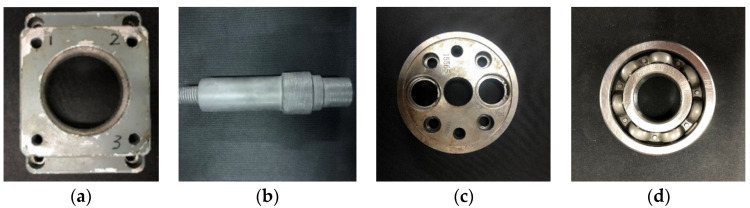
Scanned objects: (**a**) object 1; (**b**) object 2; (**c**) object 3; (**d**) object 4.

**Figure 10 sensors-22-02666-f010:**
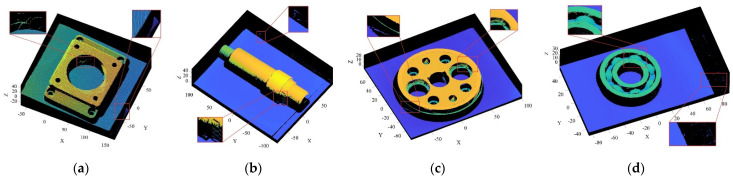
Point cloud models scanned by laser: (**a**) object 1; (**b**) object 2; (**c**) object 3; (**d**) object 4.

**Figure 11 sensors-22-02666-f011:**
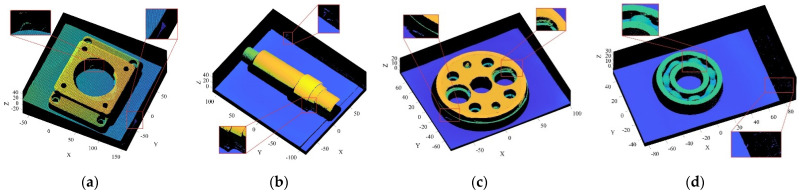
Denoising results for APSS algorithm: (**a**) object 1; (**b**) object 2; (**c**) object 3; (**d**) object 4.

**Figure 12 sensors-22-02666-f012:**
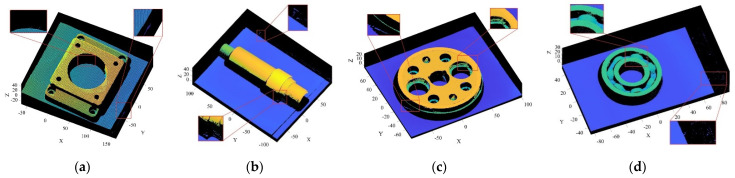
Denoising results for NLD algorithm: (**a**) object 1; (**b**) object 2; (**c**) object 3; (**d**) object 4.

**Figure 13 sensors-22-02666-f013:**
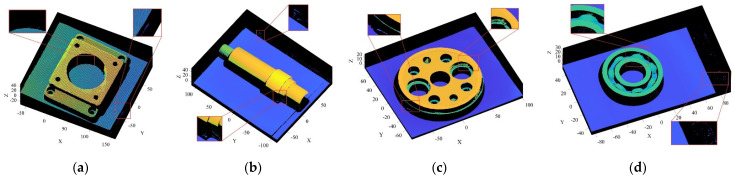
Denoising results for FGL algorithm: (**a**) object 1; (**b**) object 2; (**c**) object 3; (**d**) object 4.

**Figure 14 sensors-22-02666-f014:**
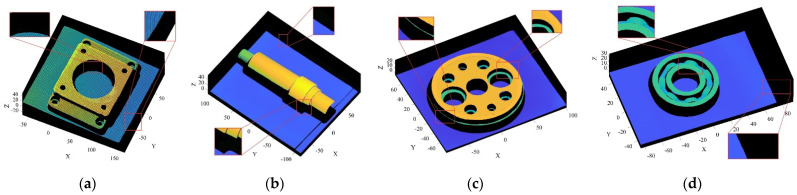
Denoising results for R-SLDM algorithm: (**a**) object 1; (**b**) object 2; (**c**) object 3; (**d**) object 4.

**Table 1 sensors-22-02666-t001:** SNR (DB) of five different denoising algorithms for Daratech models.

Noise Levels	σ = 0.01	σ = 0.02	σ = 0.03	σ = 0.04	σ = 0.05	Means
Noise	47.31	46.73	46.14	45.26	44.96	46.08
APSS	48.12	47.52	46.87	45.96	45.48	46.79
NLD	47.93	46.28	46.71	45.78	45.38	46.42
FGL	48.05	48.29	46.22	46.75	45.83	47.03
R-SLDM	49.78	48.77	48.26	47.36	46.98	**48.23**

**Table 2 sensors-22-02666-t002:** MSE of five different denoising algorithms for Daratech models.

Noise Levels	σ = 0.01	σ = 0.02	σ = 0.03	σ = 0.04	σ = 0.05	Means
Noise	0.196	0.231	0.283	0.315	0.356	0.276
APSS	0.187	0.216	0.267	0.289	0.313	0.255
WLOP	0.179	0.208	0.257	0.273	0.308	0.245
FGL	0.174	0.192	0.236	0.274	0.301	0.235
R-SLDM	0.139	0.153	0.162	0.169	0.172	**0.159**

**Table 3 sensors-22-02666-t003:** SSIM of five different denoising algorithms for Daratech models.

Noise Levels	σ = 0.01	σ = 0.02	σ = 0.03	σ = 0.04	σ = 0.05	Means
Noise	0. 944	0.921	0.913	0.895	0.856	0.906
APSS	0.954	0.936	0.967	0.949	0.913	0.944
WLOP	0.959	0.948	0.957	0.931	0.928	0.945
FGL	0.981	0.953	0.931	0.937	0.931	0.947
R-SLDM	0.979	0.983	0.962	0.961	0.952	**0.967**

**Table 4 sensors-22-02666-t004:** SNR (DB) of different denoising algorithms for four objects.

Objects	a	b	c	d	Means
Noise	45.21	47.51	48.42	44.23	46.34
APSS	46.63	48.63	48.98	45.63	47.47
NLD	46.75	47.74	48.78	46.14	47.35
FGL	46.92	47.82	48,83	47.81	47.85
R-SLDM	47.83	48.64	49.74	47.73	**48.49**

**Table 5 sensors-22-02666-t005:** MSE (×10^−3^) of different denoising algorithms for four objects.

Objects	a	b	c	d	Means
Noise	4.734	4.348	5.378	5.134	4.989
APSS	4.257	3.789	4.898	4.568	4.378
WLOP	4.191	3.695	4.788	4.414	4.272
FGL	3.933	2.978	3.764	3.789	3.616
R-SLDM	3.275	2.784	3.356	3.246	**3.165**

**Table 6 sensors-22-02666-t006:** SSIM of different denoising algorithms for four objects.

Objects	a	b	c	d	Means
Noise	0.884	0.912	0.878	0.934	0.902
APSS	0.913	0.941	0.918	0.968	0.935
WLOP	0.931	0.965	0.908	0.944	0.937
FGL	0.956	0.953	0.928	0.954	0.950
R-SLDM	0.975	0.984	0.956	0.976	**0.973**

## Data Availability

Not applicable.

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
