# Peer review of "Denoising for 3D Point Cloud Based on Regularization of a Statistical Low-Dimensional Manifold"

_sensors, 2022, doi:10.3390/s22072666_

Round 1
Reviewer 1 Report
The authors designed a de-noising algorithm for 3d point cloud based on a valid mathematical model. They also presented strong evidence that their algorithm works. I suggested publication of the paper.
Author Response
Thank you for your comments.
Reviewer 2 Report
The manuscript proposed a Denoising method for point cloud based on regularization and statistical low dimensional Manifold. The topic is relevant to the sensor journal. However the manuscript needs improvement.
That said, the work does look technically promising, and I highly encourage the authors to make the changes suggested.
Introduction:
1. The authors must include more recent methods of the state of the art, for example:
C. Dinesh, G. Cheung, I. V. Bajic, and C. Yang, "Fast 3D point cloud denoising via bipartite graph approximation & total variation", arXiv:1804.10831 [eess], Apr. 2018.
J. Zeng, G. Cheung, M. Ng, J. Pang, and C. Yang, "3D point cloud denoising using graph laplacian regularization of a low dimensional manifold model", IEEE Transactions on Image Processing, vol. 29, pp. 3474-3489, 2020, doi: 10.1109/TIP.2019.2961429.
C. Duan, S. Chen, and J. Kovacevic, "Weighted multi-projection: 3d point cloud denoising with tangent 519 planes", in 2018 IEEE Global Conference on Signal and Information Processing (GlobalSIP), Nov. 2018, pp. 520 725-729, doi: 10.1109/GlobalSIP.2018.8646331
Wei Hu; Xiang Gao; Gene Cheung; Zongming Guo. Feature Graph Learning for 3D Point Cloud Denoising. IEEE Transactions on Signal P...Volume: 68, Page(s): 2841 - 2856, DOI:10.1109/TSP.2020.2978617. 06 March 2020
Results
Authors should compare the proposed method with some of the suggested methods, some of which have public source code.
Methods:
The writing and presentation of the mathematical notation is confusing for the reader, for example in line 74 it is said that u ∈ Rmxn, but it is not said initially that it is u, the same for equation (2) line 84, it is not define neither b nor Φ, u is said to be an image, but we are in the context of point clouds, why the assertion that u is an image? The same thing happens on line 129 where it is written "M is discretely sampled to construct a ?, and to describe the low-dimensional", but it is not specified that it is ?. Line 93, ni ∈ R3, but why not says that ni is a point in R3?
Please review whole section, I suggest you check out Osher's article and how they describe and introduce mathematical notation.
I would also suggest improving the structure of the article, which could have the following sections: introduction, previous work, proposed method, results and discussion, and finally the conclusions.
Contribution:
1. technical novelty should be clarified.
2. The motivation is not clear. Please specify the importance of the proposed solution.
3. How this work is different from state of the art and which is it real contribution.
4. In the conclusion section, the limitations of the proposed method must be discussed by the authors and to related need of further work.
5. Consider the conclusion from the reader's perspective. At the end of a paper, a reader wants to know how to benefit from the work you accomplished in your paper.
6. There's no in-deep analysis to tell us why the proposed algorithm is the best.
Reviewer 3 Report
Rewrite the abstract that what is the purpose of doing this work.
Highlight the contributions of the proposed work in introduction.
How this work is different from existing one?
What is the novelty?
Don't write conclusion in points.
Recent references of 2021 & 2022 can be added of relevant journal.
Reviewer 4 Report
This manuscript proposed a 3D point cloud denoising using low demensional manifold model with regularization. My comments are as follows.
- I would suggest improving the readability of the manuscript by adding flowchart, diagram, or figures to illustrate the algorithm.
- in the example result figure 11, it's hart to tell the difference between different method, and the original image seems has no significant noise. Please add some more explanation.
- Since there are a lot of simple denoising method e.g. fast non-local mean, filtering, anisotropic diffusion, etc. along with deep learning-based denoising method, it would be more solid to compare the proposed method with traditional and state-of-the-art methods.
- I recommend more evaluation metrics such as SSIM or so.
Round 2
Reviewer 2 Report
The authors have done a good job, taking into account the suggestions made by the reviewers
Reviewer 3 Report
Accepted in its current form.